# Neurotransmitter Release Site Replenishment and Presynaptic Plasticity

**DOI:** 10.3390/ijms22010327

**Published:** 2020-12-30

**Authors:** Sumiko Mochida

**Affiliations:** Department of Physiology, Tokyo Medical University, Tokyo 160-8402, Japan; mochida@tokyo-med.ac.jp

**Keywords:** action potential, active zone, synaptic vesicle, presynaptic proteins, myosin, dynamin, Ca^2+^ channels, Ca^2+^ sensors, presynaptic plasticity

## Abstract

An action potential (AP) triggers neurotransmitter release from synaptic vesicles (SVs) docking to a specialized release site of presynaptic plasma membrane, the active zone (AZ). The AP simultaneously controls the release site replenishment with SV for sustainable synaptic transmission in response to incoming neuronal signals. Although many studies have suggested that the replenishment time is relatively slow, recent studies exploring high speed resolution have revealed SV dynamics with milliseconds timescale after an AP. Accurate regulation is conferred by proteins sensing Ca^2+^ entering through voltage-gated Ca^2+^ channels opened by an AP. This review summarizes how millisecond Ca^2+^ dynamics activate multiple protein cascades for control of the release site replenishment with release-ready SVs that underlie presynaptic short-term plasticity.

## 1. Introduction

Ultrastructural studies have demonstrated that some SVs are in contact with the presynaptic plasma membrane in the AZ [1,2], and docked vesicles are thought to represent fusion-competent vesicles [3,4,5]. After an AP, electrophysiology and electron microscopy studies have indicated that recovery of the docked and readily releasable vesicle pools (RRP) is about 3 s [3,4,6]. However, recent studies suggest that SV replenishment comprises several kinetically and molecularly distinct steps, some of which may occur on extremely fast timescales [7,8] and are reversible with Ca^2+^ control [5,9]. The ‘zap-and-freeze’ method, after generating a single AP high-pressure freezing at defined time points, was developed by Watanabe and co-workers [10]. Applying this approach to mouse hippocampal neurons in culture, they characterized the spatial and temporal organization of SV fusion sites following a single AP and demonstrated that multiple vesicles fuse within the same AZ, and ~40% of docked vesicles are lost immediately after stimulation due to fusion and, potentially, undocking. These are then fully replaced by newly docked vesicles within 14 ms. This transient docking requires residual Ca^2+^ in the AZ and only lasts for 100 ms or less. Watanabe and co-workers suggested the sequence of rapid redocking, and subsequent slow undocking may underlie facilitation.

Presynaptic Ca^2+^ concentration changes are dynamic functions in space and time, with wide fluctuations associated with different rates of neuronal activity. Thus, regulation of transmitter release, including that of SV fusion sites, involves reactions of multiple Ca^2+^-dependent proteins, each operating over a specific time window following AP-triggered Ca^2+^ dynamics. To study Ca^2+^-dependent molecular control of SV dynamics following neuronal activity, I have established a useful mammalian synapse model based on sympathetic superior cervical ganglion neurons in long-term culture. Using the synapse model, we have electrophysiologically studied the roles of presynaptic proteins in control of neuronal activity-regulated SV states in the transmitter release site and in that of SV recycling to the site [11]. 

This review introduces, at first, recent findings on temporal regulation of SV states within 100 ms of a single AP, and then our findings on millisecond Ca^2+^ dynamics-dependent transmitter release site replenishment with release-ready SVs that involves multiple protein cascades, such as phosphorylation of AZ proteins, activation of myosin motors and that of key proteins linking exocytosis and endocytosis. These protein reactions controlled by Ca^2+^ sensors underlie the presynaptic short-term plasticity. Furthermore, I would introduce an important regulation of Ca^2+^ elevation in the AZ. After a single AP, Ca^2+^ channel opening is temporally fine-controlled by the AP-induced Ca^2+^ dynamics via Ca^2+^ sensors bound to the Ca^2+^ channel. Regulation of Ca^2+^ elevation significantly controls the state and replenishment of SVs and contributes to presynaptic plasticity. 

## 2. SV Dynamics in the AZ after a Single AP

SV fusion takes place within milliseconds of an AP [12]. We see Ω-shaped membrane structure corresponding to a SV fusion with plasma membrane in electron micrographic image of freeze-substituted nerve terminals taken 5 ms after stimulation [13]. SVs fuse at specialized presynaptic membrane domain, the AZ [14], enriched with many proteins, including Ca^2+^ channels. In most synapses, the AZ is organized into one or more release sites, individual units are thought to fuse a single SV [15]. Using the ‘zap-and-freeze’ method, Watanabe and co-workers succeeded morphologically to capture SV dynamics with millisecond timescale triggered by a single AP [10]. They demonstrated in cultured mouse hippocampal neurons that, at 5 ms, 18% of the synaptic profiles exhibited exocytic pits in the AZ, in contrast, only 2% and 1% of the synaptic profiles exhibited pits in unstimulated cells and in the presence of tetrodotoxin, an AP blocker reagent, respectively. 

Whether single- or multivesicular release predominates from an AZ has long been debated [16]. To define it, Watanabe and co-workers reconstructed entire AZs from serial sections taken at 5 ms after stimulation and suggested that multivesicular release is prominent in cultured hippocampal neurons. Multivesicular release was augmented by increasing extracellular Ca^2+^. At low (1.2 mM) Ca^2+^ concentrations, neighboring SVs fuse simultaneously; pits were often within ~100 nm of each other, suggesting that release sites are coupled in a common Ca^2+^ nanodomain [17]. In contrast, at high (4 mM) Ca^2+^ concentrations, release sites act independently in a Ca^2+^ microdomain [18]; the median distance between pits was roughly like the distance between docked SVs. 

Docked SVs are often referred to as release-ready vesicles [2]. Watanabe and co-workers performed morphometry on synaptic profiles frozen at 5, 8, 11, and 14 ms after an AP, and showed striking evidence that docking is not a stable state and that vesicles can stay docked, fuse, or potentially undock with an AP (Figure 1). AP induces two processes of SV fusion—synchronous fusion occurs within 5 ms throughout the AZ, and then, at 11 ms, asynchronous fusion follows in the center of the AZ. During synchronous fusion, docked SVs across all synaptic profiles are reduced by ~40%, whereas SVs close to the membrane but not docked (between 6 and 10 nm) slightly increase, possibly undocked vesicles from the AZ but still associated with release site. Such SVs are proposed to exist in a “loose state”, with SNAREs, synaptotagmin-1, and Munc13 still engaged [19]. During asynchronous fusion, docked vesicles are not further depleted despite ongoing fusion, suggesting that SVs are recruited during this process. At 14 ms, docked vesicles are fully restored to pre-stimulus levels. This fast recovery during vesicle fusion is Ca^2+^-dependent and temporary and appears to be lost within 100 ms. This transient docking is followed by a slower docking process that requires 3–10 s. Watanabe and co-workers concluded that SVs at the AZ exhibit lively dynamics between docked and undocked states within milliseconds of an AP, supporting presynaptic short-term plasticity [20].

Watanabe and co-workers demonstrated that a single AP initiates SV fuse, and the fused SVs fully collapse into the plasma membrane by 11 ms [10] and the plasma membrane is recruited by ultrafast endocytosis at the edge of AZ [3] (see Section 5). In contrast, during bursting neuronal activity, “kiss-and run” fusion/fission SV dynamics is observed with morphological [21] and imaging studies [22,23]. The majority of these SVs in hippocampal neurons undergo another exocytosis event within 120 nm of their original fusion site and a second exocytosis event within 10 s of the first fusion event [23], allowing neurons to maintain neurotransmitter release during bursting activity.

## 3. AZ Serves as a Platform for SV Docking

### 3.1. SV Docking and Ca Channel Cluster in AZ

AZ serves as a platform for linking SVs to Ca^2+^ channels through protein complexes (Figure 2). The distance between docked SVs and Ca channels that determinates transmitter release probability differs with AZ architecture. In developing calyx of Held from P7 to P14 the distance between SV and Ca^2+^ channel cluster decrease from 30 nm to 20 nm, and the number of Ca_V_2.1 channel per cluster and the cluster area increase with development. However, the density of Ca_V_2.1 channel remains similar [24]. The mature calyx of Held synapse has numbers of bouton-like swellings on stalks of the nerve terminals that show lower release probability than that of stalks. Wong and colleagues, measuring the distance of fluorescently tagged-Ca^2+^ channels and SVs coupling using a Ca^2+^ chelator, explored that larger clusters of Ca^2+^ channels with tighter coupling distance to SVs elevate release probability in stalks, while smaller clusters with looser coupling distance lower release probability in swellings [25].

As described in next Section 3.2, Ca^2+^ channels are installed in the AZ membrane by AZ protein(s), which mediates linkage of docked SVs and Ca^2+^ channels at the release site. Stanley proposes that a single Ca^2+^ channel domain gates SV fusion at a fast synapse [26]. She and co-workers developed an in vitro SV pull-down assay [27] and presented evidence that Ca_V_2.1 channel or the mid-region of its C-terminal captures an SV [28], and Ca_V_2.2 channel or the distal third of its C-terminal [29] captures an SV as far as 100 nm from the AZ region [30,31]. They hypothesized that one or more additional linker molecule(s) lock the captured SV within its Ca^2+^ sensor to trap Ca^2+^ at the channel mouth [31].

### 3.2. AZ Proteins Regulate SV Docking Close to Ca^2+^ Channel

Watanabe and co-workers suggested that AP-triggered SV dynamics between docked and undocked states in the AZ is likely controlled by SNAREs, synaptotagmin-1, and Munc13 [19]. Munc13 is recently proposed that its action underlies activity-dependent augmentation of SV pool size [32]. The AZ is a highly organized structure with proteins that serves as a platform for SV exocytosis, mediated by SNARE proteins complex, nearby Ca^2+^ channels (Figure 2). This arrangement establishes the tight spatial organization required for fast SV fusion upon Ca^2+^ entry, and sets the synaptic strength [33]. Although the full molecular composition of the AZ is unclear, many AZ proteins have been identified; these include Munc13, RIM, RIM-BP, Bassoon, Piccolo, Liprin-α, and CAST [34,35,36,37,38,39,40,41]. These are all relatively large proteins with significant domain structures that interact with each other, forming a large macromolecular complex (Figure 2) [42]. Among AZ proteins, RIM [38] seems to be a key protein for SV dynamics [43,44]. RIM (Rab3-interacting molecules) mediates linkage of docked SV and Ca^2+^ channel at the release site [45] (Figure 2). Removing RIM lacks SV docking and slows exocytosis speed [46]. RIM interacts with other AZ proteins and the C-terminal sequences of Ca^2+^ channel [47,48] (Figure 2). A molecular complex consisting of RIM and C-terminal tails of the Ca^2+^ channel that determines recruitment of Ca^2+^ channels to the AZ includes CAST (cytomatrix at the active zone-associated structural protein) [37]. Bruchpilot, a *Drosophila* ortholog of CAST, is required for Ca^2+^ channel clustering and AZ structural integrity [49,50], suggesting a potential function for CAST in the mammalian AZ [51]. Indeed, CAST controls Ca^2+^ channel density [52] and AZ size [53,54]. We have demonstrated that disruption of CAST interaction with RIM or Bassoon impairs synaptic transmission [38]. Removing RIM lacks SV docking [46]. In addition, impairment of Bassoon slows down SV reloading [55]. These observations suggest that the CAST complex, including RIM and Bassoon plays fundamental roles in setting the synaptic strength. Thus, in the following section, I summarize our observations, studied with a model synapse formed between sympathetic superior cervical ganglion neurons in long culture, that CAST plays a role in determination of AZ architecture and in control of SV replenishment within milliseconds of the AP [56].

### 3.3. CAST Determines AZ Architecture and Electrical Signal

Overexpression of CAST in sympathetic presynaptic neurons with microinjection of the recombinant DNA increased the size of the AZ [56]. 4-fold greater than the normal AZ. Interestingly, the AZ expressing phosphomimetic-CAST^S45D^-mutant showed lower density of other AZ proteins, such as Bassoon. These changes in the AZ suggest that CAST regulates the AZ architecture in control of AZ protein distribution, and thereby determines the efficacy of transmitter release. Indeed, CAST overexpression prolonged the excitatory postsynaptic potential (EPSP) duration and increased the integral by 1.7-fold, due to increase in the number of SVs in the RRP [57], whereas the EPSP peak amplitude was unchanged. In contrast, the phosphomimetic-CAST^S45D^ expression reduced the peak amplitude and the integral of EPSP, and SV number in the RRP [56]. These results suggest that CAST controls the number of docked SVs regulating AZ size, but not the SV release probability, and that the CAST^S45^ phosphorylation decreases the number of release-ready SVs. Immunocytochemical experimentally, increase in phosphorylated CAST^S45^ in presynaptic terminals was observed with presynaptic APs burst or membrane depolarization with high K^+^ [56]. 

### 3.4. CAST Phosphorylation Controls Rapid Replenishment of Release-Ready SVs after an AP 

Phosphorylation of AZ proteins induced by APs burst lasts minutes [58]. This is not the case of CAST phosphorylation. A single AP appears to phosphorylate CAST, and the time window for the CAST phosphorylation after an AP can be estimated by the paired-AP protocol (Figure 3A,C) [56]. The first AP induces CAST phosphorylation, leading to decrease in the paired-EPSP recovery that lasts 30–120 ms after the first AP. In contrary, phosphonegative-CAST^S45A^-mutant reduces the paired-EPSP depression (<200 ms) [59,60], suggesting phosphorylated CAST^S45^ down-regulates SV reloading shortly after an AP, but not over longer (200 ms) time courses. This means that an AP activates a kinase in the AZ to phosphorylate CAST^S45^ that negatively controls release-ready SV replenishment. The possible kinase is an AZ-associated serine/threonine kinase SAD-B that phosphorylates CAST^S45^ [56]. SAD-B also phosphorylates RIM [61]. Thus, phosphorylation of key molecules in the AZ proteins complex by SAD-B may contribute to this ultrafast regulation of SV dynamics. It is likely that this down-regulation is important for sharping the transmitter release and saving release-ready SVs for an incoming AP. Indeed, after the burst of APs, rapid reloading of release-ready SVs from the reserve pool [59,60] or the replacement state vesicle pool [8] was accelerated with overexpression of CAST or the phosphonegative-CAST^S45A^ expression, while it was slowed with the phosphomimetic-CAST^S45D^ expression [56] (Figure 3B). CAST^S45^, and possibly be RIM phosphorylation, decreases the number of release-ready SVs by braking SV docking during and after intense synaptic activity.

Acute deletion of CAST by short hairpin RNAs (shRNAs) in the cultured sympathetic neuron synapse caused no significant reduction in EPSP amplitude, however, significantly delayed the rate of the fast reloading of release-ready SVs following depletion of the RRP with APs burst [56]. Indeed, the paired-EPSP depression was facilitated by CAST deletion [56]. These results indicate that the inactivation of CAST by phosphorylation, shortly <200 ms after an AP, brakes transmitter release for the next AP arriving at the presynaptic terminal. This presynaptic regulation is reasonable for relatively low firing rate and slow signal conduction of sympathetic postsynaptic neurons.

## 4. SV Transport to Release Sites

Watanabe and co-workers suggest that undocked SVs in the AZ just after an AP corresponds to SVs in “replacement sites” [8], reported in a cerebellar “simple synapse” composed of a single AZ where SVs reversibly translocate from “replacement sites” to “docking sites” within milliseconds of an AP [7,8]. The rapid SV translocation from “replacement sites” to “docking sites” is actin- and myosin-dependent [7]. We have studied and accumulated evidence that myosin isoforms function in SV transport in presynaptic terminals of long-term cultured sympathetic neurons [62,63,64]. Thus, after introducing studies on synaptic myosin in the following section, I summarize our observations with a model synapse that myosin VI plays a role in release-ready SV replenishment within milliseconds of an AP, while IIB supports the SV replenishment during and after APs firing [64]. 

### 4.1. Myosin in Synapses

Myosins are actin-based cytoskeletal motors using energy derived from ATP hydrolysis [65]. Myosin II [66], V [67], and VI [68] have specific roles for synapse function [69,70,71] and several forms of synaptic plasticity [72,73]. At synapse, their roles are diverse, including actin cytoskeleton dynamics in dendritic spines and powering of synaptic cargo transport [71]. 

Although the role of myosin in growth cone motility, but not SV mobilization, had been reported [64,74], in 1994, we electrophysiologically demonstrated a role of presynaptic myosin in the regulation of neurotransmitter release from a model synapse formed between sympathetic neurons in long culture: Presynaptic APs activate myosin light chain kinase, and the resultant actin-myosin II interaction is involved in mobilization of SVs to maintain neurotransmitter release [62]. Later, direct evidence of myosin II mobilizing SVs during synaptic firing was shown by an imaging study of hippocampal neurons [70]. Myosin II involvement in SV motility and synaptic transmission was also demonstrated at the *Drosophila melanogaster* neuromuscular junction [75,76], where myosin VI is also required for SV localization, short-term facilitation [77], and SV dynamics [78]. In brain synapses, myosin VI postsynaptically transports receptors [65,69,79], while myosin II presynaptically mobilizes SVs near the release site [7,80,81]. 

The readily releasable SV cluster (i.e., the RRP) is thought to be filled up by motor proteins from a larger SV cluster, called the recycling pool or reserve pool, during sustained neural signals of APs [80,81]. However, electrophysiological studies on the calyx of Held synapses [82] and the cerebellar synapses of the parallel fiber and the molecular layer interneuron revealed myosin II control on SV dynamics in the RRP [7]. In these synapses, the RRP consist of two pools, a fast- and a slowly releasing pool [8,83]. In response to APs burst, myosin II converts slowly releasing SVs to fast releasing ones [82] with a rapid rate constant [7]. The fast- and slow-releasing SVs pools are likely corresponding to docked SVs and undocked SVs observed morphologically after a single AP [10]. However, these two SVs pools converted by myosin II are defined under APs train: The latency distribution exhibits a single fast component at train onset but contains both a fast and a slow component later in the train [8]. Thus, in the next sections, I summarize myosin activation with diverse AP firing patterns for release-ready SV replenishment in sympathetic presynaptic neurons [64].

### 4.2. Myosin II and VI Replenish Release-Ready SVs within Milliseconds of an AP

Among three myosin II isoforms A, B, and C, myosin IIB, and myosin VI are specifically expressed in presynaptic terminals of sympathetic superior cervical ganglion neurons in long-term culture [63,64]. Myosin IIB is Ca^2+^/calmodulin (CaM)-dependently activated by myosin light chain kinase, while VI is directly activated by Ca^2+^/CaM. Participation of IIB and VI in the RRP replenishment was monitored by the recovering kinetics of release-ready SVs with diverse AP firing patterns within presynaptic neurons acutely reduced with microinjection of myosin IIB or VI small interfering RNA (siRNA) [62,64]. 

The question, whether a single AP can activate myosin II or not, can be answered by applying the paired-AP protocol (Figure 3A,C and Figure 4A) [64]. As described in Section 3.4, the sympathetic neurons show synaptic depression at short intervals of APs within 120 ms [84,85,86]. Knockdown of myosin IIB with its siRNA did not change the paired EPSP size, suggesting that a single AP is not sufficient to activate myosin IIB. In contrast, knockdown of myosin VI potentiated the decrease in the paired EPSP with ≥50 ms intervals. This means that a single AP can activate myosin VI within 50 ms, but not myosin IIB, and completes reloading of release-ready SVs into the neurotransmitter release site within 120 ms.

The questions, how many APs can activate myosin IIB and how long does it take place, can be answered by applying repetitive AP. APs at 10 Hz revealed that the time window of myosin IIB action is short, within hundreds of milliseconds (Figure 4B) [64]. Following APs at 10 Hz, the third EPSP was sensitive to myosin IIB knockdown, suggesting that myosin IIB takes 200 ms or needs two APs for the release-ready SV replenishment. In contrast, the second EPSP was sensitive to VI knockdown, suggesting that myosin VI resupplies SVs to the release site within 100 ms of an AP. Activation of myosin IIB, but not VI, needs more frequent APs > 0.1 Hz [64].

The site where myosin acts is predictable as that myosin IIB and VI contribute to local SV mobilization to the release site from two pieces of evidence. With the myosin IIB or VI knockdown, (1) number of release-ready SVs (≈50 SVs) was similar to that of control (see the first EPSP in Figure 4B), (2) it was reduced severely following repetitive APs (Figure 4B), but the size of the RRP reduction was small, 7% or 14% less than those of controls calculated with the EPSP wave form [64].

### 4.3. Myosin IIB and VI Replenish Release-Ready SVs through Distinct Pathways

We have accumulated evidence that replenishment of release sites with release-ready SVs is achieved through distinct pathways: After APs burst, reloading of release-ready SVs shows two phases, fast and slow, due to different molecular contribution [59,60,64,85,86]. The contribution of myosin IIB and VI in the fast and slow reloading was examined with their knockdown, and the SV kinetics was estimated by the recovery rate, applying the same depletion-recovery protocol as described in Section 3.4 and Figure 3B [64]. After APs burst at 5 Hz for 4 min, time constant for the fast recovery is τ = 8 s, and that for the slow recovery is τ = 4–5 min. Myosin IIB knockdown moderated the fast recovery to τ = 14 s, but not the slow recovery, whereas VI knockdown accelerated the fast recovery rate to τ = 2 s and delayed the slow recovery. The double knockdown significantly delayed both fast and slow recovery (Figure 4C). Thus, myosin IIB and VI replenish release-ready SVs through the fast and slow SV resupply pathways during and after repetitive APs (Figure 5).

## 5. Endocytosis Regulates Release-Ready SV Restoration

### 5.1. Dynamin Action Contributes to Restore Release-Ready SVs

Most forms of endocytosis require a key protein dynamin [87]. Dynamin mediates vesicle fission from the presynaptic terminal membrane [88]. Three dynamin isoforms exist [89,90]: Dynamin 1 and 3 are highly expressed in brain, whereas dynamin 2 is ubiquitous [89,91]. Ablation studies suggested that, in central neurons, dynamin 1 mediates fast SV recycling with high-frequency neuronal activity [90,92], while dynamin 2 mediates SV recycling after synaptic firing [90]. 

Watanabe and co-workers demonstrated in cultured hippocampal synapses that ultrafast endocytosis, depending on dynamin activity and actin-cytoskeleton polymerization, is required for SV docking at the release sites of AZ [3,4]. With dynamin inhibitor dynasore treatment [93], repetitive stimuli (3–100 at 20 Hz) accumulated strings of endocytic vesicles attached to the plasma membrane at the edges of AZ [3]. The vesicle in strings (≈53 nm diameter) was larger than SVs (~40 nm) but smaller than large endocytic vesicles (~80 nm) formed after a stimulus, reflecting reduced fusion during repetitive stimuli. Thus, the ultrafast endocytosis removes membrane from the surface during synaptic firing and is required for rapid restoration of the SV fusion sites. It should be noted that a single AP-triggered quantal release at mammalian end-plate measured in low Ca^2+^ or high Mg^2+^ showed single and multiple vesicles release and less failure [94], suggesting membrane retrieval can be achieved even in low level of Ca^2+^ rise in the AZ.

As described above, Watanabe and co-workers demonstrated that a single AP triggers ultrafast endocytosis at the edges of AZ; at 20 ms after a single light pulse stimulus, large endocytic vesicles are formed at the edges of AZ [10]. Accordingly, in presynaptic superior cervical ganglion neurons, rapidly activated dynamin by a single AP contributes to restore release-ready SVs. EPSPs evoked by paired-AP in dynamin siRNA-microinjected presynaptic neurons indicates that, within 20 ms of an AP, dynamin 2 and 3 are activated and restore release-ready SVs, although dynamin 2 less contribute to the restoration. In contrast, dynamin 1 takes ≥50 ms for the release-ready SV restoration. It should be noted a morphological observation that internalized endocytic vesicles are seen ≥100 ms after stimulation [4], although endocytic vesicle can be formed at 20 ms. This indicates that formation of endocytic vesicles at the edges of AZ is a time limiting step for the release-ready SVs restoration. Clearance and repriming of release sites is previously proposed as the rate-limiting step for the SV exocytosis [83]. 

During the burst of APs in the sympathetic neurons, even low frequency such as 10 Hz, knockdown of dynamin 1, but not that of dynamin 3, increases failure of transmitter release, depending on the rate of APs. Dynamin 2 less contribute to release-ready SV restoration during APs burst. Like evidence observed in different synapses, dynamin 1 mediated SV recycling guarantees release-ready SVs restoration in response to sustained neuronal activities in the presynaptic sympathetic neurons. After a long burst of APs, the fast and the slow recovery of release-ready SVs (see Figure 3B, Figure 4C) are mediated through distinct SV recycling pathways activated by dynamin 1 and 3 [85]. Double knockdown of dynamin 1 or 3 and myosin IIB or VI confirmed the contribution of individual isoforms to the fast and slow components (Figure 4C). In conclusion, after a long burst of APs, myosin IIB acts in the fast loading of release-ready SVs close to the release site where is restored by dynamin 1-mediated membrane recycling pathway, while VI supports the slow SV loading through dynamin 3-mediated pathway. Both pathways reload a common SV pool in the release site, depending on Ca^2+^ signal accompanying neural activity (Figure 5) [85].

### 5.2. Ca^2+^ Sensors Links Exocytosis to Endocytosis

Ultrafast endocytosis can be triggered by a single AP [10], suggesting a possible mediation of low affinity Ca^2+^ sensors that links exocytosis to endocytosis. Inactivation of synaptotagmin 1 function impairs SV endocytosis in the *Drosophila* neuromuscular junction [95]. Kavalai et al. proved the role of synaptotagmin 1 in single- as well as multivesicle endocytic events in rat hippocampal neurons [96]. In addition, a Ca^2+^ sensor for the asynchronous release [97,98,99], synaptotagmin 7 is responsible for slowed endocytosis seen under synaptotagmin 1 loss-of-function [96]. 

As described above, in rat sympathetic presynaptic neurons, dynamin 3 is involved in ultrafast endocytosis within 20 ms of an AP. For the dynamin 3-mediated ultrafast endocytosis, synaptotagmin 7 acts mainly as the Ca^2+^ sensor (Tanifuji and Mochida, unpublished data). Knockdown of synaptotagmin 7 with its siRNA shows potentiated depression of paired EPSP similar extent to that with dynamin 3 knockdown. Double knockdown of synaptotagmin 7 and dynamin 3 did not show a further additive decrease in paired EPSP seen with synaptotagmin 7 or dynamin 3 knockdown. Notably, the double knockdown induced failure of paired EPSP seen in 50% of paired-AP application with the interval 20 ms.

On the contrary, in response to repetitive neuronal activity, during and after a burst of APs, the dynamin 1-mediated fast vesicle membrane recycling (Figure 4C) guarantees rapid restoration of release-ready SVs [85]. This is delayed similarly with synaptotagmin 1, 2, or 7 knockdowns, while dynamin 3-mediated slow synaptic vesicle recycling (Figure 4C) is delayed with synaptotagmin 1 or 2 knockdowns. These results indicate that both synaptotagmin 1 and 2 act to trigger endocytosis under physiological frequency neuronal activity, while synaptotagmin 7 is mainly required for the ultrafast endocytosis after a single shot of neuronal activity. In contrast to the specific localization of synaptotagmin 1 and 2 to SVs, synaptotagmin 7 is more concentrated on the presynaptic plasma membrane or other internal membranes, but not SVs [100,101]. Thus, synaptotagmin 1, 2 and 7 undergo completely different membrane recycling pathways.

## 6. Presynaptic Short-Term Plasticity

### 6.1. Ca^2+^-Binding Proteins Regulate Transient SV Dynamics after an AP

Watanabe and co-workers suggested that time-dependent SV dynamics of undocking and docking at the AZ triggered by a single AP may underlie presynaptic short-term plasticity [10]. AP induces a massive reduction of docked vesicle, even in synapses that do not exhibit fusion, suggesting that Ca^2+^ drives docked vesicles into an undocked state, leading to synaptic depression. In fact, in cultured sympathetic neurons, paired-AP depression with the interval of 15 ms is reduced by fast Ca^2+^ chelation with BAPTA [86]. In hippocampal presynaptic neurons, docking levels of SVs are fully restored within 14 ms [10]. Ca^2+^ sensor synaptotagmin 1 likely mediate this transient docking [5]. However, the transient docking is not stable and declines within 100 ms along with falling calcium levels [10], suggesting that the unstable SV docking sate within 100 ms is implicated in the presynaptic short-term facilitation mediated by several Ca^2+^-sensor molecules.

The time-dependent SV dynamics described above is regulated by Ca^2+^ sensor molecules which bind to raised Ca^2+^ after a single AP. Synaptotagmin 1, 2, and 9 serve for the synchronous transmitter release [102,103,104], while synaptotagmin 7 that binds slowly to Ca^2+^ via its C_2_A domain [105] mediates asynchronous transmitter release [106]. In addition to synaptotagmin 7, several molecules, such as SNARE protein, VAMP4 [107], and SNAP23 [108], and a Ca^2+^ sensor protein DOC2 [109], are implicated in asynchronous release. Synaptotagmin 7 also mediates paired-pulse facilitation and synaptic facilitation during repetitive firing [99]. Synaptotagmin 7 is more concentrated on the presynaptic plasma membrane, perhaps bridging SV and plasma membrane may enhance fast SV fusion in response to repetitive firing [106], in addition to membrane clearance by its linkage of exocytosis and endocytosis as discussed in Section 5.2. 

Synaptic depression is often related to Ca^2+^-dependent phosphorylation of synaptic proteins. Long-term depression in cerebellar [110] and hippocampal synapses [111], for example, involve presynaptic and postsynaptic activity of the Ca^2+^/CaM-activated protein phosphatase. In addition, low-frequency depression of transmitter release caused by stimulation at 0.2 Hz, at crayfish phasic synapses, requires presynaptic phosphatase calcineurin activated by Ca^2+^/calpain [112]. 

Ca^2+^ buffers such as calbindin, parvalbumin, and related Ca^2+^-binding proteins l control synaptic strength [113,114,115] and Ca^2+^ homeostasis in the presynaptic termina [116]. Calbindin, a rapid Ca^2+^ buffer [117], alters short-term synaptic facilitation [118], whereas parvalbumin, a slow Ca^2+^ buffer [119], controls decay rate of short-term plasticity [120]. Differential expression of these Ca^2+^-dependent regulatory proteins may provide a means of cell-type-specific regulation of short-term synaptic plasticity that reflects SV states in the AZ.

### 6.2. Ca^2+^ Channel Modulation Regulates Presynaptic Short-Term Plasticity

#### 6.2.1. Presynaptic Ca^2+^ Channels

Presynaptic short-term plasticity may be a combination of multi-molecular mechanisms, in addition to the C2-domain Ca^2+^ sensor synaptotagmin 1, 7, and 9 action and Ca^2+^ buffers. Modulation of Ca^2+^ current by presynaptic Ca^2+^ channel regulation, responding to millisecond Ca^2+^ dynamics after neuronal firing, is also an important mechanism for presynaptic short-term plasticity [121]. In presynaptic neurons, P/Q-type, N-type, and R-type Ca^2+^ currents [122] are activated with APs [123]. P/Q-type and N-type Ca^2+^ currents mediate synchronous and asynchronous SV fusion, at most fast synapses [124,125,126]. Thus, activity of the Ca^2+^ channels controls neurotransmitter release efficacy. The Ca^2+^ channel proteins consists of a pore-forming α_1_ subunit associated with β, α_2_-δ and possibly γ subunits [127]. The α1 subunit incorporates the conduction pore, the voltage sensors and gating apparatus, and the channel regulation sites by second messengers, drugs, and toxins. The large intracellular segments serve as a signaling platform for Ca^2+^-dependent regulation of synaptic transmission [101,108,111,118,125,128]. 

Ca^2+^ channel α1 subunits encoded by ten distinct genes in mammals are divided into three subfamilies [125,129,130]. The Ca_V_2 subfamily members Ca_V_2.1, Ca_V_2.2, and Ca_V_2.3 channels conduct Ca^2+^ currents classified as P/Q-type, N-type, and R-type, respectively [125,126,129,130]. At most fast synapse in the brain, presynaptic Ca_V_2 channels are expressed diversely. At the calyx of Held synapse, for example, a combination of P/Q- and N-type presynaptic currents in young mice show activity-dependent facilitation [131,132]. The Ca^2+^ current facilitation and synaptic facilitation are lost by Ca_V_2.1 channel deletion [131,132,133]. The remaining N-type Ca^2+^ currents do not support synaptic facilitation and are less efficient for synaptic transmission [133]. These observations suggest that neuronal firing controls diversely expressed presynaptic Ca^2+^ channels for modulation of synaptic transmission and synaptic plasticity. 

#### 6.2.2. Short-Term Plasticity with CaM or Neuron-Specific Ca^2+^-Sensor Proteins Binding to the Ca_V_2.1 Channel

The Ca_V_2.1 channel mediates facilitation of the presynaptic Ca^2+^ current, leading to synaptic facilitation [131,132,133]. Catterall and his colleagues found that repetitively generated Ca^2+^ currents increase and then decrease due to the channel modulation by Ca^2+^ elevation [134]. With Ca^2+^ elevation the Ca_V_2.1 channel bind to CaM [125,134,135,136,137] or to the related neuron-specific Ca^2+^-binding proteins, calcium-binding protein 1 (CaBP1), visinin-like protein-2 (VILIP-2) [138,139,140] and neuronal calcium sensor-1 (NCS-1) [141]. CaM with four EF-hand Ca^2+^-binding motifs binds to the α_1_ subunit at the IQ-like motif (IM) and the nearby CaM-binding domain (CBD) in the C-terminal [137]. At the same site as CaM binding, neuron-specific Ca^2+^-sensor proteins CaBP1, VILIP-2, and NCS-1 modulate Ca_V_2.1 channel activity. CaBP1, highly expressed in the brain and retina [142], binding to the CBD, causes Ca_V_2.1 channel inactivation [140]. VILIP-2, highly expressed in the neocortex and hippocampus [143], binding to both IM and CBD, increases Ca^2+^-dependent Ca_V_2.1 channel facilitation but inhibits Ca^2+^-dependent inactivation [139]. NCS-1, binding to both IM and CBD, reduces Ca^2+^-dependent Ca_V_2.1 channel inactivation [141]. 

The physiological role of these Ca^2+^-sensors-mediated presynaptic Ca_V_2.1 channel modulation was explored by expressing the α1 subunit derived from the brain Ca_V_2.1 channel that functionally generates P/Q type currents with other endogenous subunits in long-term cultured sympathetic neurons [121] where Ca_V_2.2 channels mediate synaptic transmission [144,145]. A single AP is sufficient to modulate the Ca_V_2.1 channel activity interacting with Ca^2+^ bound CaM (Ca^2+^/CaM) and changes paired EPSP [121]. Ca^2+^ binding sites of CaM, N and C lobes, sense differently increase in Ca^2+^ concentration. The N-lobe sensing rapid and higher increase in Ca^2+^ concentration [146] binds to the CBD, while the C-lobe sensing lower Ca^2+^ concentration binds to the IM. Thus, pairs of APs induced synaptic depression and facilitation with varied stimulation intervals (Figure 6A). AP with <50 ms interval decreased EPSP, while AP with 50–100 ms interval facilitated EPSP. Declining in Ca^2+^ elevation after the first AP causes temporal regulation of the Ca_V_2.1 channel activity mediated by Ca^2+^/CaM, resulting in a change in the efficacy of SV fusion triggered by following AP (Figure 6B). 

APs burst with short interval induces presynaptic short-term plasticity with the Ca^2+^/CaM-dependent Ca_V_2.1 channel regulation. Ca^2+^/CaM binding to the CBD controls negatively the SV fusion efficacy, whereas Ca^2+^/CaM binding to the IM controls it positively. APs burst with a higher frequency over 20 Hz shapes the time course of short-term presynaptic plasticity by Ca^2+^/CaM-mediated Ca_V_2.1 channel inactivation (Figure 6C), suggesting that it controls the peak timing of synaptic facilitation during APs burst as well as the steady-state level of synaptic depression at the end of the APs burst.

#### 6.2.3. Temporal Regulation of Release Efficacy by Ca^2+^-Sensor Proteins

Presynaptic AP opens a Ca^2+^ channel and creates a steep gradient of Ca^2+^ elevation in the AZ. Within the transient Ca^2+^ elevation, each Ca^2+^-sensor proteins, having a different affinity and binding speed to Ca^2+^ [146], determines the timing of the Ca_V_2.1 channel modulation. Ca^2+^ concentrations for CaM, CaBP1 and VILIP-2 binding to Ca^2+^ are 5–10 μM, 2.5 μM and ~1 μM, respectively [147]. After an AP, CaM modulates Ca_V_2.1 channel activity shortly after Ca^2+^ entry and lasts 100 ms (Figure 6A,B), while NCS-1 acts much shorter, 30–50 ms [141]. CaBP1 and VILIP-2 actions start later and last longer, 50–150 ms and 50–250 ms, respectively, than CaM effects [148]. The divergent actions of CaM and neuron-specific Ca^2+^-binding proteins on the Ca_V_2.1 channel causes various short-term plasticity at different synapses [149]. Thus, a temporal regulation of the Ca_V_2.1 channel activity by each Ca^2+^-sensor would substantially control transmitter release efficacy in response to non-stop neuronal signaling [150].

## 7. Conclusions

SV fusion takes place within milliseconds after the firing of an AP [12]. SV dynamics in the AZ, demonstrated morphologically by a method called ‘zap-and-freeze’ at defined time points after a single AP, revealed that millisecond Ca^2+^ dynamics after an AP control SV states in the AZ, not only in the synchronous fusion, but also undocking, redocking, and asynchronous fusion states [10]. The undocking and redocking states are not stable and contribute to presynaptic short-term plasticity. Our electrophysiological study, following changes in EPSP after an AP, showed that release-ready SVs are controlled by the millisecond Ca^2+^ dynamics [86] that activate multiple protein cascades via Ca^2+^-sensor molecules, including AZ proteins complex, motor proteins, and endocytic proteins [56,64,85] (Figure 7). Furthermore, the millisecond Ca^2+^ dynamics also control presynaptic Ca^2+^ channels activity, by Ca^2+^ bound CaM and neuron-specific Ca^2+^-binding proteins [121,141,148,151], to modulate Ca^2+^ elevation with incoming APs. These multiple protein cascades activated by a single AP should be activated during and after bursts of APs, resulting in presynaptic short-term plasticity. In conclusion, release-ready SVs in the AZ are controlled by Ca^2+^ dynamics with each AP, and the SV fusion efficacy changes sustainably in response to neuronal activity.

## Figures and Tables

**Figure 1 ijms-22-00327-f001:**
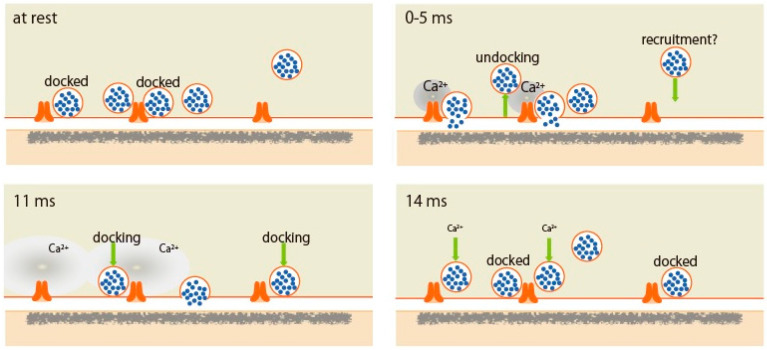
Scheme of synaptic vesicle dynamics in the active zone demonstrated by the ‘zap-and-freeze’ method. (At rest) Synaptic vesicles (SVs) beneath the active zone (AZ) are transit between docked and undocked states. Docked SVs are ready to fuse in response to action potential (AP). (0–5 ms) Synchronous SV fusion, often of multiple SVs, starts within hundreds of microseconds of an AP. Fused SVs collapse into the plasma membrane by 11 ms. (11 ms) From 5 to 11 ms, SV fusion, toward the center of the AZ, is asynchronized. SVs start to be recruited. (14 ms) SVs are docked to fully replace the used SV for fusion. These docked SVs reduces within 100 ms. Reproduced from Kusick et al., 2020 [10].

**Figure 2 ijms-22-00327-f002:**
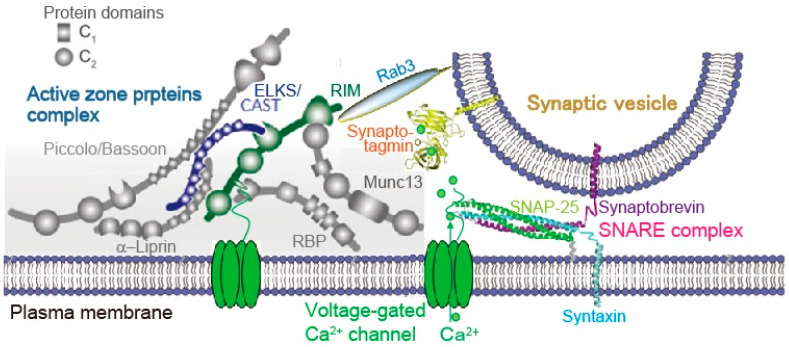
Active zone proteins. The active zone is a highly organized structure that docks synaptic vesicles (SVs) close to fusion machinery proteins (SNAREs) and Ca^2+^ channels. This establishes the tight spatial organization required for fast exocytosis upon Ca^2+^ entry, and it provides molecular machinery to set and regulate synaptic strength. Reproduced from Wang S et al., 2016 [26].

**Figure 3 ijms-22-00327-f003:**
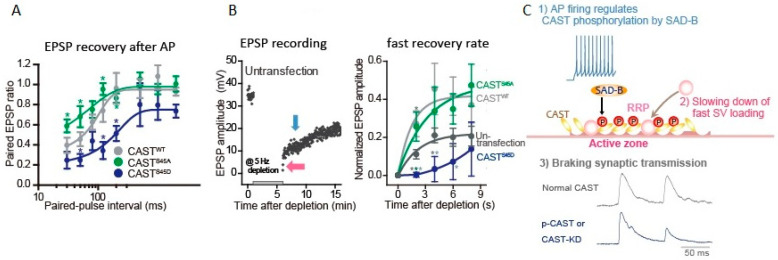
Active zone protein CAST plays a role in synaptic vesicle loading. The presynaptic neurons were transfected with CAST^S45^ DNA, 2 days before EPSP recording. (**A**) Phosphonegative-CAST^S45A^ relieved the second EPSP reduction, while phosphomimetic-CAST^S45D^ potentiated the reduction. Paired EPSP ratio was plotted against paired-AP interval. (Bar: SEM. * *p* < 0.05; unpaired student’s *t*-test. *green, CAST^S45A^ vs. CAST^WT^; * navy, CAST^S45D^ vs. CAST^WT^) (**B**) EPSP recovery from depletion of SVs. A 5-min train of APs at 5 Hz was applied, as indicated, for SV depletion. EPSP amplitude recovered with two different rates: fast (pink arrow) and slow phases (blue arrow) (left). The fast recovery rate is estimated by an exponential fit to the increase in EPSP amplitude from 0 to 8 s after the depletion (right). (* *p* < 0.05, ** *p* < 0.01, *** *p* < 0.001.; unpaired student’s *t*-test. *dark gray, CAST^S45D^ or CAST^WT^ vs. un-transfection; *green, CAST^S45D^ vs. CAST^S45A^; *gray, CAST^S45D^ vs. CAST^WT^) (**C**) Cascaded reactions of CAST phosphorylation. Adapted from Mochida et al., 2016 [56].

**Figure 4 ijms-22-00327-f004:**
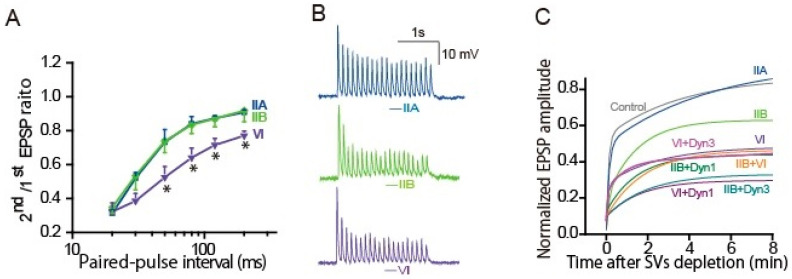
Myosin IIB and VI support release-ready SV restoration. Presynaptic neurons were transfected with control, myosin IIA-, IIB-, or VI-siRNA 2–3 days before the EPSP recordings. (**A**) Myosin VI, but not IIB, knockdown reduced release-ready SVs after AP-evoked transmitter release. The averaged paired-EPSP ratio was plotted against the paired-AP interval. (Bar: SEM * *p* < 0.05; Bonferroni post hoc test after one-way ANOVA. VI- knockdown vs. control IIA- or IIB- knockdown) (**B**) Myosin IIB and VI knockdown impaired transmitter release during APs firing at 10 Hz, resulted in a sudden reduction in third and second EPSP amplitude, respectively. (**C**) Myosin IIB and VI knockdown delayed recovery from the RRP depletion. After a 1 min control recording at 1 Hz, a 4-min AP train at 5 Hz was applied to deplete SVs. EPSP amplitudes were normalized to the mean EPSP amplitudes before the 4-min train. Recovery rate of release-ready SVs is estimated by double exponential growth fit to the increase in averaged EPSP amplitude after the depletion. +Dyn1: double knockdown of dynamin-1 and myosin; +Dyn3: double knockdown of dynamin-3 and myosin.

**Figure 5 ijms-22-00327-f005:**
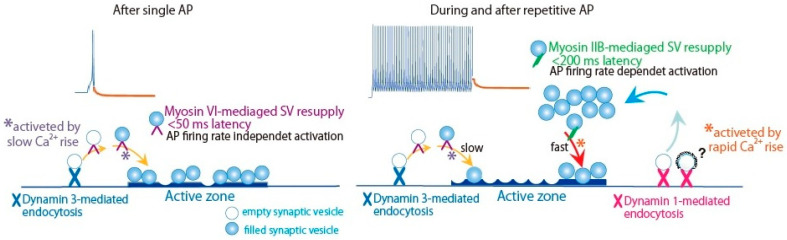
Neural activity selects Myosin IIB and VI in distinct dynamin isoform-mediated SV resupply pathways. Schematic drawing of SV resupply to the release site by myosin VI and IIB through distinct SV membrane recycling pathways mediated by dynamin isoforms. *purple, SV reloading activated by slow Ca^2+^ rise; *orange, SV reloading activated by rapid Ca^2+^ rise. ?, Clathrin-coated vesicle endocytosis is under debated [3,13]. SV clusters are classified into two pools: release-readily vesicles on the active zone membrane [57] and replacement site vesicles [8]. Adapted from Lu et al., 2009 [60]; Tanifuji et al., 2013 [85]; Mori et al., 2014 [86]; Hayashida et al., 2015 [64].

**Figure 6 ijms-22-00327-f006:**
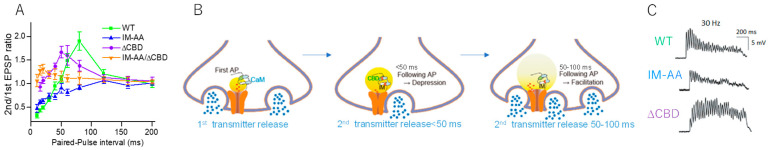
AP modulates synaptic transmission via Ca_V_2.1 channel regulation. Ca_V_2.1 channel, wild type (WT), alanine mutation at the IQ-like motif (IM), or deletion of the CaM-binding domain (ΔCBD) was expressed in presynaptic neurons and EPSPs were recorded in the presence of a blocker for native Ca_V_2.2 channels. (**A**) Paired-AP alters the second EPSP amplitude. Timing of the second AP-induced depression and facilitation of synaptic transmission. Depression was prevented by ΔCBD, while facilitation was prevented by IM-AA mutation of the pore-forming α1 subunit. (**B**) Model for Ca^2+^/CaM-dependent inactivation and facilitation of the Ca_V_2.1 channel and neurotransmitter release. (**C**) Biphasic synaptic transmission during 1-s APs at 30 Hz changed to synaptic depression by IM-AA or synaptic facilitation by ΔCBD mutation. Adapted from Mochida et al., 2008 [121].

**Figure 7 ijms-22-00327-f007:**
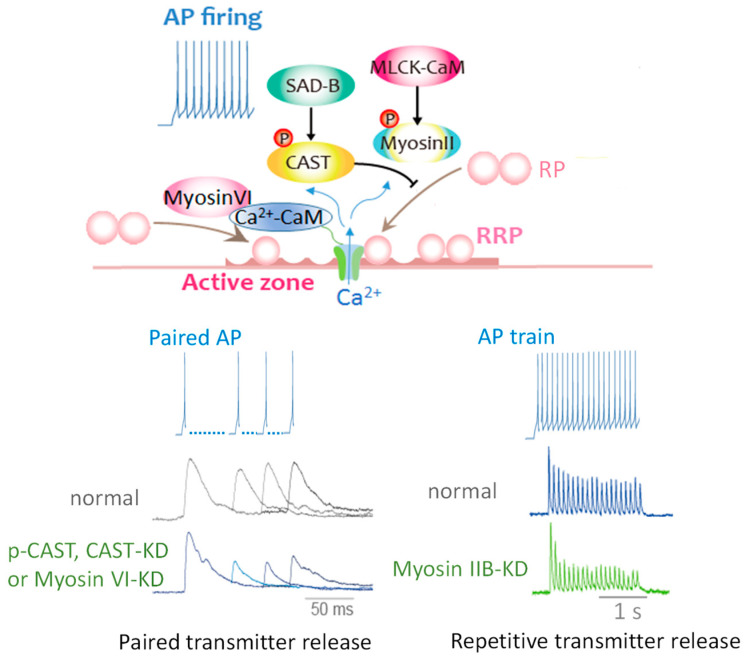
Milliseconds Ca^2+^ dynamics activates multiple proteins for controlling release-ready SVs. Myosin IIB and VI activated by Ca^2+^/CaM dependent on APs firing pattern translocate SVs to restore release-ready state SVs. In contrast, APs-induced CAST phosphorylation brakes release-ready state SV restoration. Elevation of Ca^2+^ with a single AP activates Ca^2+^/CaM/myosin VI and SAD-B at the AZ and controls the speed of SV reloading into the RRP (upper scheme), resulting in modulation of transmitter release efficacy for an incoming AP (bottom left traces). Repetitive APs control the efficacy by myosin IIB activation (upper scheme) and possibly by other proteins in the AZ for sustainable synaptic transmission (bottom right traces). All these key protein function and property of Ca^2+^ channels in the AZ regulated by AP via Ca^2+^ sensors action determine release efficacy and underlie presynaptic short-term plasticity. (RP, replacement state SV pool [8]; RRP, release-ready state SV pool [57]; KD, knockdown).

## Data Availability

Not applicable.

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
