# Peer review of "Neurotransmitter Release Site Replenishment and Presynaptic Plasticity"

_ijms, 2020, doi:10.3390/ijms22010327_

Round 1
Reviewer 1 Report
Review: Neurotransmitter release site replenishment and presynaptic plasticity
The review covers the main and recent topics in synaptic transmission on a focus of the presynaptic site. The focus on Ca2+ dynamics was helpful in pulling together the timing of vesicle fusion and Ca2+ regulation as well as the role of CAST. The relation with various model preparations was helpful, such as bruchpilot in Drosophila with CAST for mammalian preparations.
Minor:
Line 198: "Although its role….". Maybe state what "its" means just to be clear
Line 210: Generally, the RRP is referred to as " readily releasable pool" not the "release-ready" just seems odd not to use the standard terminology. (Kaeser, P.S.;Regehr, W.G. The readily releasable pool of synaptic vesicles. Curr Opin Neurobiol 2017,43: 63-70.)
Line234 other places as well:
Just using the word "AP" in the way the author uses it .... implies "APs" are occurring so there is no need to state AP "firing". Maybe just state AP. "That a single AP is not sufficient to…."
Line 241: "Following APs at 10 Hz….." would read better.
Or "Following APs occurring at 10 Hz…
The author may like to know about these references
- Low freq depression..................Silverman-Gavrila LB, Praver M, Mykles DL, Charlton MP. Calcium, calpain, and calcineurin in low-frequency depression of transmitter release. J Neurosci. 2013 Jan 30;33(5):1975-90. doi: 10.1523/JNEUROSCI.3092-12.2013. PMID: 23365236; PMCID: PMC6619111
- The review cited covers a lot (Rizzoli, S.O.;Betz, W.J. Synaptic vesicle pools. Nat Rev Neurosci2005, 6(1), 57-69.) But this review in 2014 covers some of the topics in the current manuscript which is worth citing.
- considering vesicles may recycle empty, can one be sure the vesicles are really inhibited from fusion when measuring by quantal responses?
Author Response
Responses to reviewer 1
Thank you very much for your valuable comments that help my poor English writing and knowledge of published papers. My corrections pointed by reviewer 1 are highlighted with green color in the text.
Line 198: "Although its role….". Maybe state what "its" means just to be clear => “Its role” is changed to “role of myosin”
Line 210: Generally, the RRP is referred to as " readily releasable pool" not the "release-ready" just seems odd not to use the standard terminology. (Kaeser, P.S.;Regehr, W.G. The readily releasable pool of synaptic vesicles. Curr Opin Neurobiol 2017,43: 63-70.) => "release-ready" is corrected to " readily releasable SV cluster"
Line234 other places as well:
Just using the word "AP" in the way the author uses it .... implies "APs" are occurring so there is no need to state AP "firing". Maybe just state AP. "That a single AP is not sufficient to…." => "firing" is removed throughout of the text, indicated with green highlight.
Line 241: "Following APs at 10 Hz….." would read better. => Text was changed to “Following APs at 10 Hz”
The author may like to know about these references
- Low freq depression..................Silverman-Gavrila LB, Praver M, Mykles DL, Charlton MP. Calcium, calpain, and calcineurin in low-frequency depression of transmitter release. J Neurosci. 2013 Jan 30;33(5):1975-90. doi: 10.1523/JNEUROSCI.3092-12.2013. PMID: 23365236; PMCID: PMC6619111
=>line 406: New description is added.
Synaptic depression is often related to Ca2+-dependent phosphorylation of synaptic proteins. Long-term depression in cerebellar [110] and hippocampal synapses [111], for example, involve presynaptic and postsynaptic activity of the Ca2+/CaM-activated protein phosphatase. In addition, low-frequency depression of transmitter release caused by stimulation at 0.2 Hz, at crayfish phasic synapses, requires presynaptic phosphatase calcineurin activated by Ca2+/calpain [112].
- The review cited covers a lot (Rizzoli, S.O.; Betz, W.J. Synaptic vesicle pools. Nat Rev Neurosci 2005, 6(1), 57-69.) But this review in 2014 covers some of the topics in the current manuscript which is worth citing.
=> Rizzoli, S.O., Synaptic vesicle recycling: steps and principles. Embo j 2014, 33(8), 788-822. This review is added in the list and cited as [82]
- considering vesicles may recycle empty, can one be sure the vesicles are really inhibited from fusion when measuring by quantal responses?
=> line 331: New description is added. It should be noted that a single AP-triggered quantal release at mammalian end-plate measured in low Ca2+ or high Mg2+ showed single and multiple vesicles release and less failure [94], suggesting membrane retrieval can be achieved even in low level of Ca2+ rise in the AZ.
In addition, introduction and several important findings pointed by other reviewers are added. 1) line 45-54, introduction, 2) line 100-106, the “kiss-and-run” event, 3) line 108-119, change in Ca2+ channel distribution in developing and mature calyx of Held synapses, 4) line 120-127, capture of synaptic vesicle by CaV2.1 and CaV2.2 channels.
Reviewer 2 Report
This is a highly competent review article by a leader in the synaptic transmission field. The work is well organized, and nicely illustrated, and will make a useful contribution to the field. I have two minor comments, as well as one slightly more substantial consideration.
Slightly major:
There is a series of papers from Elise Stanley's lab in Frontiers in Cellular Neuroscience (2013, 2014,2017,2018) that deals with capture and tethering of of synaptic vesicle. This work was ignored her even though it might have been relevant to section 4. This should have been included one way or another (even if you do not agree with it). Also, work on calyx of Held is mentioned, but some of the seminal work from Dr. Lu-yang Wang's group is not cited (for example Fekete et al, Nat Communications). I realize that the topic is not as much about vesicle release as about replenishment, but synaptic architecture vis a vis calcium channels is discussed here and so the author may wish to include the above works.
Minor:
In the section on calmodulin, "robe" should be "lobe"
The author uses SV as an abbreviation for synaptic vesicle. Here, plural is used incorrectly throughout the manucript - for example, if the authors want to say replenishment of synaptic vesicles, then "SVs" should be used
In contrast "Endocytosis regulates release-ready SVs restoration" is incorrect and should be "SV restoration" or alternatively, "restoration of SVs" - this type of error happens many times throughout the text. So every time you use "SVs" as an abbreviation, please make sure that it is correctly used
Author Response
Responses to reviewer 2
Thank you very much for your valuable comments that help my poor English writing and knowledge of published papers. My corrections listed by reviewer 2 are highlighted with blue color in the text.
Slightly major:
There is a series of papers from Elise Stanley's lab in Frontiers in Cellular Neuroscience (2013, 2014,2017,2018) that deals with capture and tethering of of synaptic vesicle. This work was ignored her even though it might have been relevant to section 4. This should have been included one way or another (even if you do not agree with it). Also, work on calyx of Held is mentioned, but some of the seminal work from Dr. Lu-yang Wang's group is not cited (for example Fekete et al, Nat Communications). I realize that the topic is not as much about vesicle release as about replenishment, but synaptic architecture vis a vis calcium channels is discussed here and so the author may wish to include the above works.
=> Works on calyx of Held synapse of Thakahashi’s goup and Lu-yang Wang’s group and series of papers from Stanley’s lab were summarized in the new subsection 3.1. 3.1. SV docking and Ca channel cluster in AZ
AZ serves as a platform for linking SVs to Ca2+ channels through proteins complex (Figure 2). The distance between docked SVs and Ca channels that determinates transmitter release probability differs with AZ architecture. In developing calyx of Held from P7 to P14 the distance between SV and Ca2+ channel cluster decrease from 30 nm to 20 nm, and the number of CaV2.1 channel per cluster and the cluster area increase with development, however, the density of CaV2.1 channel remains similar [24]. The mature calyx of Held synapse has numbers of bouton-like swellings on stalks of the nerve terminals that show lower release probability than that of stalks. Wong and colleagues, measuring distance of fluorescently tagged-Ca2+ and SVs coupling using Ca2+ chelator, explored that larger clusters of Ca2+ channels with tighter coupling distance to SVs elevate release probability in stalks, while smaller clusters with looser coupling distance lower release probability in swellings [25].
As describing in next 3.2 section, Ca2+ channels are installed in the AZ membrane by AZ protein(s) which mediates linkage of docked SVs and Ca2+ channels at the release site. Stanley proposes that a single Ca2+ channel domain gates SV fusion at fast synapse [26]. She and co-workers developed an in vitro SV pull-down assay [27] and presented evidence that CaV2.1 channel or the mid-region of its C-terminal captures an SV [28], and CaV2.2 channels or the distal third of its C-terminal [29] capture an SV as far as 100 nm from the AZ region [30, 31]. They hypothesized that one or more additional linker molecule(s) lock the captured SV within its Ca2+ sensor trap Ca2+ at the channel mouth [31].
Minor:
In the section on calmodulin, "robe" should be "lobe" => corrected.
The author uses SV as an abbreviation for synaptic vesicle. Here, plural is used incorrectly throughout the manucript - for example, if the authors want to say replenishment of synaptic vesicles, then "SVs" should be used. In contrast "Endocytosis regulates release-ready SVs restoration" is incorrect and should be "SV restoration" or alternatively, "restoration of SVs" - this type of error happens many times throughout the text. So every time you use "SVs" as an abbreviation, please make sure that it is correctly used. => carefully corrected throughout of the text.
In addition, introduction and several important findings pointed by other reviewers are added. 1) line 45-54, introduction, 2) line 100-106, the “kiss-and-run” event, 3) line 331-333, the “quantal release” event, 4) line 406-410, presynaptic depression
Reviewer 3 Report
The present review is well written and clear. Several very accurate details are reported by the author about the synaptic vesicles machinery. Few points require attention in my opinion:
Introduction section is not well focused, and can better explain the literature concerning the main arguments reported in the rest of the paper
Figures are too detailed reporting several plots and graphs. I think this is redundant for a review and make difficult sometime to follow the text
Author report several data about synapse model based on sympathetic superior cervical ganglion neurons. Is some case these data seems overrepresented with respect the rest of the results.
Pg 2 line 46, reference reported seems wrong
Author Response
Responses to reviewer 3
Thank you very much for your valuable comments. My corrections pointed by reviewer 3 are highlighted with pink color in the text.
Introduction section is not well focused, and can better explain the literature concerning the main arguments reported in the rest of the paper => Thank you for your pointing out. I added description explaining it.
This review introduces, at first, recent findings on temporal regulation of SV states within 100 ms of a single AP, and then our findings on millisecond Ca2+ dynamics dependent transmitter release site replenishment with release-ready SVs that involves multiple protein cascades, such as phosphorylation of zone proteins, activation of myosin motors and that of key proteins linking exocytosis and endocytosis. These protein reactions controlled by Ca2+ sensors underlie the presynaptic short-term plasticity. Furthermore, I would introduce an important regulation of Ca2+ elevation in the AZ. After a single AP, Ca2+ channel opening is temporally fine-controlled by the AP-induced Ca2+ dynamics via Ca2+ sensors bound to the Ca2+ channel. Regulation of Ca2+ elevation significantly controls state and replenishment of SVs and contributes presynaptic plasticity.
Figures are too detailed reporting several plots and graphs. I think this is redundant for a review and make difficult sometime to follow the text => Thank you for your pointing out. I divided Figure 2 to Figure 2 and 3, and Figure 3 to Figure 4 and Figure 5, hoping easier understanding of figure contents.
Author report several data about synapse model based on sympathetic superior cervical ganglion neurons. Is some case these data seems overrepresented with respect the rest of the results. => Several important findings pointed by other reviewers are added. 1) line 100-106, the “kiss-and-run” event, 2) line 108-119, change in Ca2+ channel distribution in developing and mature calyx of Held synapses, 3) line 120-127, capture of synaptic vesicle by CaV2.1 and CaV2.2 channels, 4) line 331-333, the “quantal release” event, 5) line 406-410, presynaptic depression.
Pg 2 line 46, reference reported seems wrong => corrected to Ma, H.; Mochida, S. A cholinergic model synapse to elucidate protein function at presynaptic terminals. Neurosci Res 2007, 57(4), 491-8.